# Ameliorated Antibacterial and Antioxidant Properties by *Trichoderma harzianum* Mediated Green Synthesis of Silver Nanoparticles

**DOI:** 10.3390/biom11040535

**Published:** 2021-04-04

**Authors:** Narasimhamurthy Konappa, Arakere C. Udayashankar, Nirmaladevi Dhamodaran, Soumya Krishnamurthy, Shubha Jagannath, Fazilath Uzma, Chamanahalli Kyathegowda Pradeep, Savitha De Britto, Srinivas Chowdappa, Sudisha Jogaiah

**Affiliations:** 1Department of Microbiology and Biotechnology, Jnana Bharathi Campus, Bangalore University, Bengaluru 560 056, Karnataka, India; n.murthy10@gmail.com (N.K.); shubha.jagannath@gmail.com (S.J.); faziuzma@gmail.com (F.U.); 2Department of Studies in Biotechnology, University of Mysore, Manasagangotri, Mysore 570 006, Karnataka, India; ac.uday@gmail.com (A.C.U.); pradeep77.gowda@gmail.com (C.K.P.); 3Department of Microbiology, Ramaiah College of Arts, Science and Commerce, Bangalore 560 054, Karnataka, India; nirmaladevi1012@gmail.com; 4Department of Microbiology, Field Marshal K. M. Cariappa College, A Constituent College of Mangalore University, Madikeri 571 201, Karnataka, India; soumyanukrish@gmail.com; 5Laboratory of Plant Healthcare and Diagnostics, PG Department of Biotechnology and Microbiology, Karnatak University, Dharwad 580 003, Karnataka, India; savitha.debritto51@gmail.com; 6Division of Biological Sciences, School of Science and Technology, The University of Goroka, Goroka 441, Papua New Guinea

**Keywords:** silver nanoparticles, bioactive metabolites, antibacterial activity, MIC, antioxidant activity, *T. harzianum* filtrate

## Abstract

Biosynthesis of silver nanoparticles using beneficial *Trichoderma harzianum* is a simple, eco-friendly and cost-effective route. Secondary metabolites secreted by *T. harzianum* act as capping and reducing agents that can offer constancy and can contribute to biological activity. The present study aimed to synthesize silver nanoparticles using *T. harzianum* cell filtrate and investigate different bioactive metabolites based on LC-MS/MS analysis. The synthesized silver nanoparticles (AgNPs) from *T. harzianum* were characterized by ultraviolet–visible spectrophotometry, Fourier transform infrared spectrometry (FT-IR), energy-dispersive spectroscopy (EDS), dynamic light scattering (DLS), X-ray powder diffraction (XRD) and scanning electron microscopy (SEM). The surface plasmon resonance of synthesized particles formed a peak centered near 438 nm. The DLS study determined the average size of AgNPs to be 21.49 nm. The average size of AgNPs was measured to be 72 nm by SEM. The cubic crystal structure from XRD analysis confirmed the synthesized particles as silver nanoparticles. The AgNPs exhibited remarkable antioxidant properties, as determined by DPPH and ferric reducing antioxidant power (FRAP) assay. The AgNPs also exhibited broad-spectrum antibacterial activity against two Gram-positive bacteria (*S. aureus* and *B. subtilis*) and two Gram-negative bacteria (*E. coli* and *R. solanacearum*). The minimum inhibitory concentration (MIC) of AgNPs towards bacterial growth was evaluated. The antibacterial activity of AgNPs was further confirmed by fluorescence microscopy and SEM analysis.

## 1. Introduction

Nanotechnology has evolved as an interesting part of research because of its ability to produce nanoparticles (NPs) possessing uniformity and special properties that make them valuable in optical sensors, drug delivery, catalysis, adsorption, water treatment and nanomedicine [1]. These NPs can be produced from different chemical, physical and biological processes. The chemical and physical approaches for the synthesis of nanoparticles are technically difficult and expensive. Chemical reduction, electrochemical and pyrolysis-based methods for the production of NPs are toxic, non-eco-friendly and difficult to perform. The requirement of strong reducing agents and toxic chemicals such as borohydride and hydrazine derivatives can increase the cost of production and result in hazardous wastes being released into the environment [2].

Biosynthetic methods have been used as a substitute for chemical and physical methods. The NPs derived from gold, silver and platinum are reported to have medical and pharmaceutical applications and are well documented to have important usages in magnetics, electronics, information storage and optoelectronics [3,4]. The biosynthesis of silver nanoparticles (AgNPs) has been proven by many researchers to be the best alternative when compared with physical and chemical methods of nanoparticle synthesis, as it is amore eco-friendly and cost-effective route [5]. The AgNPs are the most significant NPs used worldwide as they have several applications in clothing, dentistry, photography, coating of surgical devices, food industry, catalysis, optics, electronics, mirrors, prostheses, textiles, wound dressing, cosmetics and agriculture [6,7]. AgNPs synthesized by green methods are described to have various biological functions, making them useful in antidiabetic, anti-inflammatory, anticancer, antimicrobial, antiplasmodial, vector control and sensor applications, and also act as catalysts [8,9].

The ever-increasing problem of antibiotic resistance in pathogenic microorganisms challenges the research community to develop cost-effective processes for synthesizing novel antimicrobial agents [10]. Silver NPs cause the destruction of microorganisms by attacking and damaging their negatively charged cell walls. Further disruption of membrane permeability and deactivation of enzymes leads to cell lysis and death [11]. The activity of AgNPs depends on the monovalent ionic silver (Ag^+^), which enters into the target microbial cells and prevents microbial growth by suppressing the activity of respiratory enzymes and electron transport mechanisms [12]. It has also been found that the AgNPs affect the cellular membrane permeability [13].

Microorganisms have been investigated as potential biofactories for nanoparticle biosynthesis [14]. Current research efforts focus on the synthesis of NPs using microbes such as algae [15], fungi [2], yeast [16] and bacteria [17] or using the byproducts of microbial metabolism, which are active reducing and stabilizing agents. Among the various microorganisms, fungal synthesis of AgNPs is simple and effective; however, the process parameters need to be optimized accordingly to attain the desired monodispersity, stability, biocompatibility and other critical properties of the particles [2,18]. Among the various fungi used for biosynthesis of AgNPs are *Fusarium oxysporum* [19], *Fusarium semitectum* [19], *Aspergillus* sp. [20], *Trichoderma harzianum* [3,21], *Beauveria bassiana* [22], *Penicillium* sp. [23], *Cladosporium* sp. [24] and basidiomycetes [7]. In this study, we have investigated the use of the fungus *Trichoderma harzianum* (MK611661) in the synthesis of extracellular AgNPs. The bioactive metabolites present in *T. harzianum* culture filtrate were identified using LC-MS/MS. Further, AgNPs were synthesized from *T. harzianum* filtrate using AgNO_3_, and the characterization of synthesized AgNPs was based on microscopic and spectroscopic methods, including ultraviolet–visible spectrophotometry, Fourier transform infrared spectrometry (FT-IR), energy-dispersive spectroscopy (EDS), X-ray powder diffraction (XRD) and scanning electron microscopy (SEM). Further, their potential antioxidant and antimicrobial activities were evaluated.

## 2. Materials and Methods

### 2.1. Isolation of Trichoderma harzianum and Biomass Preparation

*Trichoderma harzianum* was isolated from rhizosphere soil on potato dextrose agar (PDA) medium at 28 °C. The isolated fungus was identified morphologically by lactophenol cotton blue mounting, and molecular identification was based on the internal transcribed spacer (ITS) sequencing. Based on the sequence, the fungus was identified as *T. harzianum*, and the sequence was deposited in National Center for Biotechnology Information (NCBI) under GenBank accession number MK611661.Mycelial disks (5 mm) of *T. harzianum* culture were inoculated into 100 mL of potato dextrose broth (PDB) medium in 250 mL Erlenmeyer flasks and incubated at 25 ± 2 °C for 5 days on a rotary shaker at 150 rpm. After 5 days of incubation, the fungal mycelium mass was separated by filtration using Whatman No. 1 filter paper. Following harvest, the biomass was washed with distilled water to remove any media components. Later, 25 g of the fungal mycelium (wet weight) was suspended in 100 mL of Milli-Q water and incubated at 25 ± 2 °C on a rotary shaker at 150 rpm for 72 h. The fungal mycelium was filtered and the cell-free filtrate was collected for subsequent experiments. The culture filtrate of *T. harzianum* was extracted with ethyl acetate (EtOAc) 3 times with a final 1:1 ratio. The combined organic fraction was dried and evaporated under reduced pressure at 35 °C and used for analysis of bioactive metabolites by LC-MS/MS.

### 2.2. Analysis of Bioactive Metabolites Present in T. harzianum Filtrate by LC-MS/MS

The chemical constituents from culture filtrate were determined using LC-MS/MS. First, 50 mg of *T. harzianum* culture filtrate extract was suspended in 2 mL of methanol and filtered through 0.22 µm nylon membrane prior to injection. HPLC was coupled with a Q-TOF mass spectrometer fitted with an ESI source. HPLC column Phenomene x 5 μ C8, (150 × 2 mm i.d.) was used for the analysis. The solvents were delivered at a total flow rate of 0.1 mL/min and run by isocratic elution. The MS spectra were acquired in the positive ion mode. The temperature of the drying gas (N2) was 350 °C, the gas flow rate was6 mL/min and the nebulizing pressure (N_2_) was25 psi. A 20 μL volume of fungal extract was injected onto the analytical column for analysis. The mass fragmentations were identified by using a spectrum database for organic compounds. The analytical LC/MS experiment was performed using a TSQ Quantum Access MAX Triple-Stage Quadrupole Mass Spectrometer. Waters Mass Lynx and Target Lynx software were used for data acquisition and data processing, respectively. The MS analysis was performed using ESI in the positive mode. The MS parameters were curtain gas 10, gas1 20 and gas 20, needle voltage 5000 V and declustering potential 100 V. TOF was operated between 50 and 1500 *m*/*z* with low mass resolution of 4.7 and high mass resolution of 15.

### 2.3. Biosynthesis of Silver Nanoparticles (AgNPs) 

For the synthesis of AgNPs, the culture filtrate was mixed with 1 mM silver nitrate solution (AgNO_3_) in the ratio of 1:9 (*v*/*v*), and the reaction mixture was incubated at 25 °C and 100 rpm overnight (to avoid photoactivation of AgNO_3_). The change in color of the solution from yellowish-brown to dark brown indicated the reduction of silver nitrate to silver ions. The dark brown solution was subjected to centrifugation at 15,000 rpm for 25 min, the supernatant was discarded and the pellet was washed 5–6 times with sterile distilled water. The AgNPs obtained were dried at 60 °C for 24 h and then used for characterization studies. The filtrate without AgNO_3_ was used as negative control.

### 2.4. Characterization of Synthesized Silver Nanoparticles (AgNPs)

The maximum absorbance of AgNPs obtained was determined by spectral scan in the range between 200 and 800 nm using a UV–Vis spectrophotometer (Hitachi, U-2800). The reduction of pure silver ions synthesized by fungal filtrate was observed by measuring the UV–Vis spectrum of the reaction mixture. The analysis of AgNPs with Fourier transform infrared spectrometry (FT-IR) was performed by scanning in the spectral range 400–4000 cm^−1^ at a resolution of 4 cm^−1^ (Perkin Elmer Spectrum 1000). FT-IR spectra in solid phase were recorded as potassium bromide pellets to detect the possible functional groups in the fungal filtrate responsible for the reduction of ions and the capping agents responsible for the stability of nanoparticles. The energy-dispersive spectroscopy (EDS) assay was conducted using 0.2 g of AgNO_3_ crystals to detect the presence of silver ions in the samples (Hitachi Noran System 7, USA). The DLS analysis was conducted to check the size and dispersal pattern of biosynthesized AgNPs existing in solution (Microtrac /FLEX 11.0.0.2). Scanning electron microscopy (SEM) analysis was carried out using a tiny film of AgNPs placed on carbon-coated copper grid film and dried using a mercury lamp for 5 min. The morphological structure obtained from the biosynthesized AgNPs was determined (Hitachi, S-3400N, Tokyo, Japan). The X-ray powder diffraction (XRD) patterns of synthesized AgNPs were detected using a X-ray powder diffractometer (Rigaku Desktop Miniflex II) with Cu Kα radiation (λ = 1.5406 A°) as the energy source. The diffracted intensities were recorded at 2*θ* angles from 10–80°. The location of the highest peak was compared with standard libraries to detect crystal-like phases. The size and nature of biosynthesized nanoparticles were obtained by XRD. The size of the NPs was determined by the Debye–Sherrer equation given as follows:D = Kλ/*β*cos*θ*(1)
where λ is the X-ray wavelength, D is the particle size (nm), *β* is the full line width at half maximum (FWHM) elevation of the important peak, K is the shape factor and *θ* is the refractive (Bragg) angle.

### 2.5. Determination of Antioxidant Activities 

#### 2.5.1. 2,2-Diphenyl-1-picryl-hydrazyl-hydrate (DPPH) Scavenging Activity Assay

The biosynthesized AgNPs and the culture filtrate were used to assess the antioxidant property by DPPH radical scavenging assay [25]. First, 1.5 mL of freshly prepared DPPH (4 mg of DPPH in 100 mL of 95% ethyl alcohol) was added to 1.5 mL of culture filtrate and AgNPs samples (0.20–1.0 mg/mL). After incubation at room temperature in the dark for 30 min, reduction of DPPH was determined spectrophotometrically at 517 nm against the blank (1.5 mL of DPPH solution and 1 mL of 95% ethanol); gallic acid was used as standard (0.2–1.0 mg/mL in 95% ethyl alcohol). The blank consisted of 1.5 mL of DPPH solution containing the filtrate. The experiments were repeated thrice. The percent activity and IC_50_ (concentration of sample needed to inhibit 50% DPPH) were determined.
Percent activity (%) = [(Ac − As) /Ac] × 100(2)
where Ac is the absorbance of control or blank and As is the absorbance of the AgNP mixture or standard.

#### 2.5.2. Ferric Reducing Antioxidant Power (FRAP) Assay

The antioxidant potential of AgNPs and the culture filtrate was analyzed by ferric reducing antioxidant power (FRAP) assay [26]. The FRAP reagent (4.5 mL) was prepared by mixing 2.5 mL of TPTZ (2,4,6-tripyridyl-striazine) solution (10 mM TPTZ in 40 mM HCl) and 20 mM FeCl_3_ in 25 mL of acetate buffer (0.3 M, pH 3.6) with 0.5 mL of test samples at different concentrations (0.2–1.0 mg/mL). Deionized water and ethanol were used as blank. The reaction mixture was incubated at 37 °C for 30 min, and the absorbance was recorded at 593 nm. A dark blue color formed as Fe^3+^–TPTZ complex was reduced to Fe^2+^–TPTZ. Freshly prepared aqueous ascorbic acid solution (0.2–1.0 mg/mL) was used as standard.

### 2.6. Determination of Antibacterial Activity by Disc Diffusion Method 

#### 2.6.1. Microbial Cultures Used for Antibacterial Activity 

*Escherichia coli* (NCIM–2256), *Staphylococcus aureus* (NCIM–2079) and *Bacillus subtilis* (NCIM–2724) obtained from the National Collection of Industrial Microorganisms (NCIM), Pune, India, were used in this study. *Ralstonia solanacearum* (RS5-KF924743) bacterium was isolated from rhizospheric soil.

#### 2.6.2. Antibacterial Activity of Synthesized AgNPs

The antibacterial efficacy of the synthesized AgNPs was evaluated by disc diffusion method using two Gram-positive bacteria, namely *S. aureus* and *B. subtilis*, and two Gram-negative bacteria, namely *E. coli* and *R. solanacearum*. Briefly, the bacterial cultures were inoculated into 5 mL of sterile nutrient broth and incubated at 37 °C until the turbidity matched the 0.5 McFarland standard. The bacterial broth was swabbed onto sterile Mueller Hinton agar (MHA) plates to obtain lawn culture. The sterile discs (6 mm) soaked overnight in 25 μL of AgNPs (0.25 mg/mL) were placed at equidistance on MHA plates and allowed to diffuse at 4 °C for 4–5 h. Streptomycin (25 μg/disc) was used as the positive control and the culture filtrate served as a negative control. The plates were then incubated at 37 °C overnight and the zone of inhibition was measured. The experiments were repeated thrice and mean values were recorded.

### 2.7. Minimum Inhibitory Concentration (MIC)

The MIC of AgNPs was determined by the dilution plate method. Various dilutions of AgNPs were prepared, ranging from 4096 to 8 μg/mL. One hundred microliters of nutrient broth was added to the micro wells of an ELISA plate, and an equal volume of AgNPs ranging from 4096 to 8 μg/mL was added to each well. Ten microliters of bacteria (1 × 10^5^CFU/mL) was added to each well before incubation at 37 °C for 24 h, and absorbance was measured at 620 nm using an ELISA plate reader. Streptomycin was used as the positive control. The MIC was also observed by the addition of 10 μL (2 mg/mL) of 2,3,5-triphenyl tetrazolium chloride (TZC), incubated at room temperature for 30 min. The lowest concentration of AgNPs that significantly inhibited the growth of bacteria in comparison with the positive control was recorded as the MIC [27]. The experiments were repeated thrice and mean values were recorded. 

### 2.8. Fluorescence Microscopy and Scanning Electron Microscopy (SEM) Analysis 

The growth of *S. aureus* (Gram-positive bacteria) and *R. solanacearum* (Gram-negative bacteria) was inhibited by AgNPs at a concentration of 0.25 mg/mL, as determined by the disc diffusion assay. Hence, 0.25 mg/mL of AgNPs was used for the detection of both dead and live cells by fluorescent microscopy. *S. aureus* and *R. solanacearum* were treated with AgNPs (0.25 mg/mL) and incubated for 24 h; then, dyes acridine orange (1 μL) and ethidium bromide (1 μL) were added before further incubation in dark condition for 10–15 min. Acridine orange invades bacterial cells and stains the nuclei green, whereas ethidium bromide invades the cell-membrane-disrupted bacteria and stains the nuclei orange [28]. Ten microliters of bacteria culture treated with fluorescent stains was placed on a slide to observe the stained nuclei under fluorescence microscope at 40× magnification. The morphological features of treated bacterial cells were further assessed by SEM.

## 3. Results

### 3.1. Analysis of Bioactive Metabolites from Fungal Filtrate by LC-MS/MS Method

The HPLC chromatogram showed several minor peaks at various retention times ranging from 0.20 to 2.20 min. The prominent peak with an area of 246,184.42 was obtained at the retention time of 1.33 min (Figure 1A). The LC-MS/MS spectra of the filtrate showed the presence of five different compounds with varied mass obtained at different retention times (Figure 1B–D). The compounds were tentatively identified based on the mass obtained from LC-MS/MS by comparison with the previously reported compounds documented in databases. The compounds were identified as 1-benzoyl-3-[(*S*)-((2*S*,4*R*,8*R*)-8-ethylquinuclidin-2-yl](6-methoxyquinolin-4-yl)methyl)thiourea (*m*/*z* 489.2323), puerarin (*m*/*z* 416.2064), genistein (*m*/*z* 432.2986), isotalatizidine (*m*/*z* 407.2975) and ginsenoside (*m*/*z* 800.5387) (Table 1). These compounds were previously reported to exhibit antimicrobial, antibacterial, antioxidant and anticancer properties.

### 3.2. Characterization of Silver Nanoparticles (AgNPs)

The green-synthesized AgNPs were characterized by ultraviolet–visible spectrophotometry, Fourier transform infrared spectrometry (FT-IR), energy-dispersive spectroscopy (EDS), dynamic light scattering (DLS), X-ray powder diffraction (XRD) and scanning electron microscopy (SEM).

#### 3.2.1. UV–Visible Spectroscopy Analysis of AgNPs

The addition of filtrate to silver nitrate (AgNO_3_) resulted in a color change from pale yellowish-brown to reddish-brown due to the formation of silver nanoparticles (AgNPs). The change in color occurred due to the excitation of surface plasmon resonance (SPR) from AgNPs. The SPR of AgNPs formed a peak centered near 438 nm (Figure 2). The absorbance of AgNPs solution was taken initially when the color of the solution was pale yellowish-brown, and it was also taken after the color of the solution turned reddish-brown. The potential synthesis mechanism included the reduction of silver ions from toxic silver cation (Ag^+^) to stable Ag° due to the capping agents that existed in the filtrate.

#### 3.2.2. Energy Dispersive Spectroscopy (EDS) Analysis of AgNPs

Energy dispersive spectroscopy (EDS) analysis confirmed the presence of AgNPs synthesized from filtrate. This analysis revealed the presence of a maximum amount of AgNPs (58.75%), followed by carbon, oxygen, chlorine, etc. The metallic AgNPs generally display an optical absorption peak at 3 keV because of surface plasmon resonance (Figure 3; Table 2).

#### 3.2.3. Dynamic Light Scattering Analysis of AgNPs

The DLS study was conducted in order to detect the particle size distribution of AgNPs synthesized from filtrate (Figure 4). The size of biosynthesized AgNPs was recorded within a range of 14.64 to 23.52 nm in diameter. The average size of the AgNPs was 21.49 nm.

#### 3.2.4. Scanning Electron Microscopy Analysis of AgNPs

The size of biosynthesized AgNPs was clearly observed by SEM study and was measured to be 72 nm (Figure 5). The NPs were found to be highly crystalline in texture. 

#### 3.2.5. Powder X-ray Diffraction Study of AgNPs

The powder X-ray diffraction pattern of biosynthesized AgNPs was documented using a Bruker D8 Advance X-ray diffractometer employing Cu Kα radiation (λ = 1.5406 Å), 40 kV to 40 mA, 2θ/θ scanning method. The information was collected for a 2θ range of 10–80° with a stage of 0.020°. The diffraction data (Figure 6) were compared with standard JCPDS, silver file No. 04-0783. The XRD patterns show peaks of AgNPs at 32.3, 38, 46 and 77° corresponding to (110), (111), (200) and (311) planes of AgNPs. The XRD study confirmed that the particles were AgNPs with face-centered cubic crystal structures. The peak in the diffraction pattern at 27° could be due to bioorganic impurities.

#### 3.2.6. Fourier Transform Infrared Spectroscopy Analysis of AgNPs

The FT-IR analysis of biosynthesized AgNPs displayed the presence of a strong peak of aromatic and metal oxide, which indicated that the aromatic secondary metabolites could have been reduced during synthesis and caused stretching at 2980 cm^−1^; the stretching of alcoholic functional group (O–H stretching) also completely disappeared. The development of metal oxide was also noticed; stretching at 399.26 cm^−1^ indicated the presence of silver oxide (Figure 7).

### 3.3. Antioxidant Activities

#### 3.3.1. DPPH Scavenging Activity of AgNPs

The percent DPPH radical scavenging activity of biosynthesized AgNPs of different concentrations (0.2–1.0 mg/mL) and filtrate in comparison with gallic acid as standard ranged from 38.6 to 64.93% (Figure 8A). The DPPH scavenging activity of the synthesized AgNPs was detected based on the change in color from violet to yellow due to the formation of diphenyl picryl hydrazine. The gallic acid (standard) exhibited maximum reducing power, with a scavenging efficacy of 83.3% at 1 mg/mL. The results of this study indicate that AgNPs exhibited maximum DPPH scavenging activity as compared with the fungal filtrate. The half-maximal inhibitory concentration (IC_50_) values were determined from the graph of regression analysis. The DPPH scavenging activity of AgNPs was recorded with an IC_50_ value of 0.79 mg/mL, and the IC_50_ value of gallic acid was 0.23 mg/mL.

#### 3.3.2. Ferric Reducing Antioxidant Power (FRAP) Assay

The ferric reducing antioxidant power of AgNPs (0.2–1.0 mg/mL) and culture filtrate increased with increasing concentration and reached a peak of 66.4% and 73.98% at 1 mg/mL, respectively (Figure 8B). The standard, ascorbic acid, exhibited maximum reducing power, with 97.49% at 1 mg/mL. The results of this study indicated that biosynthesized AgNPs exhibited higher FRAP activity when compared with filtrate.

### 3.4. Antibacterial Activity of AgNPs

The biosynthesized AgNPs exhibited broad-spectrum antibacterial activity, inhibiting all the Gram-positive and Gram-negative bacteria tested. The maximum zone of inhibition was observed against Gram-positive *Staphylococcus aureus* (14.6 mm), followed by *Bacillus subtilis* (13.86 mm). Among the Gram-negative bacteria, inhibition zones of 17.43 and 15.56 mm were exhibited against *R. solanacearum* and *Escherichia coli*, respectively (Figure 9; Table 3).

### 3.5. Minimum Inhibitory Concentration(MIC)

The MICs of biosynthesized AgNPs against Gram-positive and Gram-negative bacteria were evaluated by the broth microdilution method. The green-synthesized AgNPs exhibited MIC of 256 µg/mL against *S. aureus,* 128 µg/mL against *E. coli,* 512 µg/mL against *B. subtilis* and 64 µg/mL against *R. solanacearum* (Table 4).

### 3.6. Fluorescence Microscopy and Scanning Electron Microscopy Analysis

The fluorescence microscopy studies confirmed the antibacterial activity of AgNPs against both Gram-positive and Gram-negative bacteria. The bacterial cells without AgNP treatment were used as control, these bacteria showed green color under fluorescence microscope, indicating the presence of live bacterial cells; the bacterial cells treated with AgNPs showed red color under fluorescence microscope, and the shrinkage of bacterial cell walls, non-homologous exterior and collapse were also noticed (Figure 10). The bacterial cells treated with AgNPs were further subjected to SEM analysis to detect the exact morphological changes in the bacteria. The bacterial cells treated with AgNPs exhibited significant noticeable morphological changes, which included the disruption of the bacterial cell wall resulting in leakage of intracellular materials, and bacterial clumps were formed (Figure 11).

## 4. Discussion

The silver element and its mixtures are reported to have antimicrobial properties and have been used to store water or in the making of coins from times immemorial [35]. Silver nanoparticles exhibit effective antimicrobial activity compared to other NPs because of their great surface area and their interaction with microbial cell walls [36]. In the present study, chemical constituents in the culture filtrate of *T. harzianum* were determined using LC-MS/MS [29,30,31,32,33,34]. The details of all the compounds identified in the *T. harzianum* culture filtrate are presented in Table 1. The different bioactive metabolites belonging to the groups of flavanones, steroids, alkaloids and phospholipids were detected using high-performance liquid chromatography coupled with tandem mass spectrometry (LC-MS/MS). The compounds were identified as 1-benzoyl-3-[(*S*)-((2D*S*, 4*R*, 8*R*)-8-ethylquinuclidin-2-yl](6-methoxyquinolin-4-yl)methyl)thiourea (*m*/*z* 489.2323), puerarin (*m*/*z* 416.2064), genistein (*m*/*z* 432.2986), isotalatizidine (*m*/*z* 407.2975) and ginsenoside (*m*/*z* 800.5387). These compounds belong to the phytochemical groups of flavanones, terpenoids, steroids and alkaloids. These metabolites exhibit vast biological activities and were reported earlier to exhibit antibacterial, antimicrobial, antioxidant and anticancer properties. Puerarin is used in treating ailments such as fever, diarrhea, toxicosis and metabolic disorders. It is also reported to possess antibacterial, antioxidant and anticarcinogenic activities (Table 1). Genistein, naturally occurring in the human diet, has potent antioxidant, anticancer and antiosteoporosis effects (Table 1). Isotalatizidine and Ginsenoside have been reported to be useful as antibacterial and antifungal agents (Table 1). The previous research reports prove the efficient antimicrobial and antioxidant properties of the bioactive metabolites associated with *T. harzianum* filtrate. The fungal metabolites that are responsive for reducing and/or capping agents were subjected to the reduction of Ag^+^ ions to Ag^0^ metal for the biosynthesis and fabrication of AgNPs from *T. harzianum* filtrate [37].

In the present study, biosynthesized AgNPs from *T. harzianum* filtrate exhibited broad-spectrum antibacterial activity, inhibiting Gram-positive and Gram-negative bacteria tested. The maximum zone of inhibition was obtained against *R. solanacearum* (17.43 mm), and the minimum zone of inhibition was obtained against *B. subtilis* (13.86 mm). Remarkably, the AgNPs synthesized using *T. harzianum* filtrate were found to be more active against the Gram-negative bacteria (*R. solanacearum* and *E. coli*) than the Gram-positive bacteria (*S. aureus* and *B. subtilis*). This finding is in agreement with Liao et al. [38], who reported that Gram-negative bacteria are usually more susceptible to Ag^+^ invasion compared to Gram-positive bacteria due to the difference in their cell wall structures. *Trichoderma* spp. are of vast economic importance due to their ability to produce several important enzymes and antimicrobial metabolites and their ability to act as potent biological control agents against many plant pathogens [4,21,39,40].

Our findings are in accordance with those of Raza et al. [41], who reported that the AgNPs exhibited significant broad-spectrum antimicrobial properties and the bioactive concentration of AgNPs matched with commercially available antibiotic drugs. The antimicrobial properties of AgNPs against several pathogenic microorganisms are attributed to numerous key mechanisms [42]. The enhanced antibacterial activity of biosynthesized AgNPs is because of their identical shape, smaller size and limited size dispersal [43]. AgNPs attach to the negatively charged cell surface; modify the physical and chemical properties of the cell wall and the cell membranes; and interrupt significant functions such as osmoregulation, permeability, electron transport and respiration [44]. Following penetration, AgNPs enter into the cytoplasm of bacteria, and AgNPs can also disrupt the cellular function by interacting with enzymes and amino acids, forming reactive oxygen species (ROS) and disintegrating the bacterial DNA [38]. The colloidal consistency of the AgNPs easily captures the bacteria, thus enhancing their interaction and entry into the cell wall [45]. The silver ions interact with diverse thiol groups of peptides/enzymes and DNA, thus interrupting DNA replication of microbes, causing structural modification and the development of granules of sulfur and interrupting the synthesis of cellular proteins and enzymes essential for ATP synthesis [46].

Previous research has proved that AgNPs synthesized from the culture filtrate of filamentous fungus *T. harzianum* and evaluated for their antibacterial activities against *Staphylococcus aureus* and *Klebsiella pneumonia* showed significant inhibition of bacterial growth in a dose-dependent manner, with the Gram-negative bacterium (*K. pneumoniae*) showing higher sensitivity [47]. Chen et al. [48] reported improved antibacterial effects of AgNPs using selected surfactants against the phytopathogenic bacterium *Ralstonia solanacearum* compared to silver ions with low MIC (4.88 μg/mL) and MBC (19.5 μg/mL). The AgNPs synthesized from *T. harzianum* filtrate resulted in dose-dependent death when they were tested against larvae and pupae of the dengue vector mosquito *Aedes aegypti* [49]. The green-synthesized AgNPs from *Aspergillus niger* showed activity against *Xanthomonas citri* and *R. solanacearum* [50].

In the present study, the green-synthesized AgNPs from fungi exhibited an absorbance range of 430–448 nm by UV–Vis spectrophotometer. The strong, single and wide surface plasmon resonance (SPR) peak obtained by UV–Vis spectrum indicated a polydispersity property of AgNPs [4]. The factors influencing the surface plasmon resonance (SPR) are the dielectric medium, particle size and chemical environment [8]. The appearance of brown color was due to excitation of SPR, which exhibited maximum absorbance in the visible range of 430–460 nm for AgNPs [51].

In the present study, the FT-IR analysis displayed major peaks for AgNPs with the highest intensity at 399.26 cm^−1^ matching to metal oxide (M-O). Our results are in accordance with the findings of Banerjee et al. [52], who reported the presence of alcohol, terpenoids and carbonyl groups providing support as tough binding spots for AgNPs by FT-IR analysis. Likewise, Vahabi et al. [53] reported the FT-IR spectrum from a drop coated film of an aqueous solution incubated with *Trichoderma reesei* and reacted with *Ag*^+^ ions for 72 h. The amide bonds identified at 1650 and 1450 cm^−1^ are due to –C=O and N–H stretch vibrations present in the amide linkages of the proteins, respectively. The images of green-synthesized AgNPs obtained from SEM exhibited identical crystalline structures aggregated closely with identical shape and morphology. The crystals of AgNPs exhibit maximum absorption at 3 keV because of SPR [54]. The C and Cl accounted fora greater proportion of weight when compared to O, which accounted for the least. The stability and strength of AgNPs are greatly influenced by the nature of crystals formed [55].

In the present study, the antioxidant activity of green-synthesized AgNPs was evaluated by DPPH free radical scavenging and FRAP assay. In the DPPH scavenging activity of AgNPs, the IC_50_ value was 0.79 mg/mL. DPPH is a stable compound that becomes reduced by receiving electrons or hydrogen and thus is widely used to assess antioxidant activities. IC_50_ values are inversely associated with DPPH scavenging activity. The free radicals from metal stimulate oxidative stress; for example, reactive oxygen species (ROS) cause destruction of cell walls, DNA and mitochondria in bacteria, finally resulting in the death of the cell [56] (Figure 12). *Trichoderma* species stimulate the production of low-molecular-weight and volatile compounds such asphytoalexins, harzianopyridone, pyrones, terpenes, peptaibols and terpenoid compounds that have antibacterial activity [57]. Certain *Trichoderma* species have been known as producers of significant secondary metabolites such as plant growth regulators, antibiotics and enzymes that mostly are used to protect plants from pathogens [58]. Enzymes secreted by the *Trichoderma* species are known to have antimicrobial, anticancer and antioxidant activity [59].

## 5. Conclusions

The field of nanotechnology has been a significantly growing area of research in recent years, as it has immense applications in healthcare, industries and environmental fields. In the current study, the silver nanoparticles synthesized from *T. harzianum* culture filtrate were stable and exhibited useful bioactivities. The green-synthesized AgNPs exhibited broad-spectrum antibacterial activity against the Gram-positive and Gram-negative pathogenic bacteria tested; significant antioxidant properties were also detected. The method adopted in this study for green synthesis of AgNPs is rapid, economically viable, ecologically friendly, nontoxic and suitable for large-scale production. However, further research is required to demonstrate the other biological activities (e.g., antifungal, antidiabetic, anti-inflammatory and cytotoxic potential) and the mechanism of action.

## Figures and Tables

**Figure 1 biomolecules-11-00535-f001:**
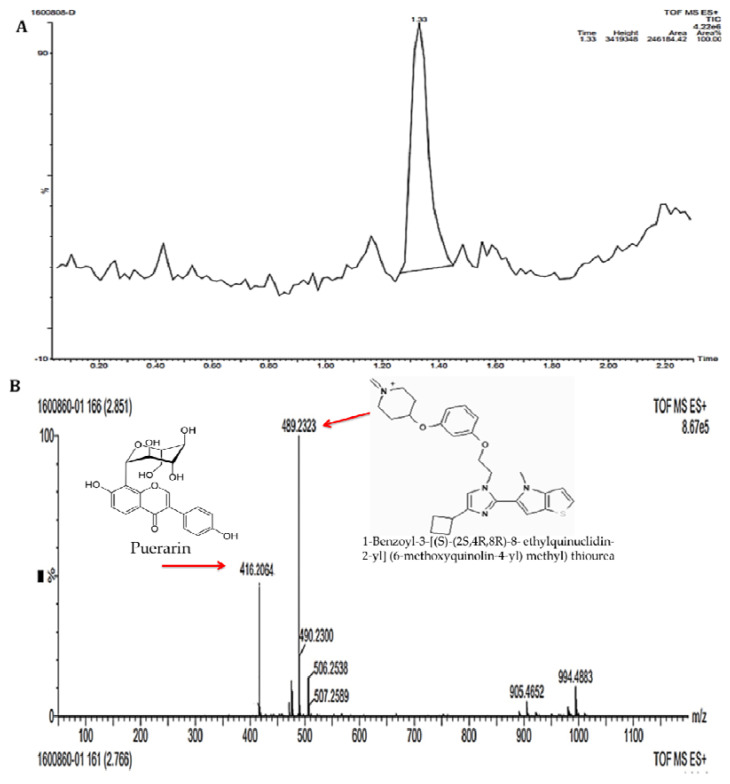
(**A**) HPLC chromatograms of the *Trichoderma harzianum* filtrate. (**B**) LC-MS/MS spectra of antibacterial compounds from *Trichoderma harzianum* filtrate: 1-benzoyl-3-[(*S*)-((2*S*,4*R*,8*R*)-8-ethylquinuclidin-2-yl](6-methoxyquinolin-4-yl)methyl) thiourea (*m*/*z* 489.2323) and puerarin (*m*/*z* 416.2064). (**C**,**D**) LC-MS/MS spectra of antimicrobial compounds from *Trichoderma harzianum* filtrate: (**C**) genistin (*m*/*z* 432.2086); (**D**) isotalatizidine (*m*/*z* 407.2975) and ginsenoside (*m*/*z* 800.5387).

**Figure 2 biomolecules-11-00535-f002:**
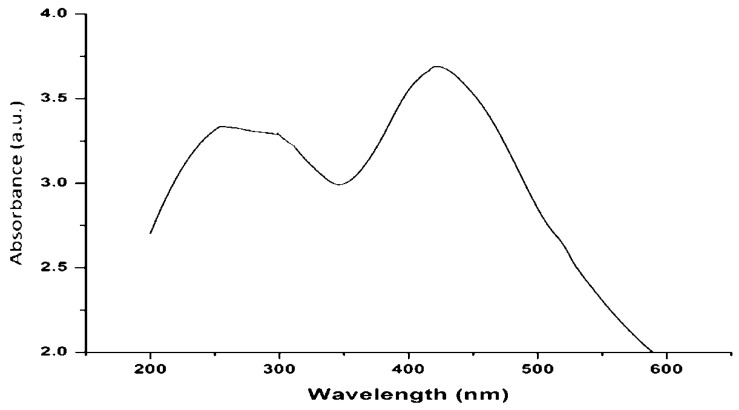
UV–visible spectra of silver nanoparticles synthesized using *Trichoderma harzianum* filtrate.

**Figure 3 biomolecules-11-00535-f003:**
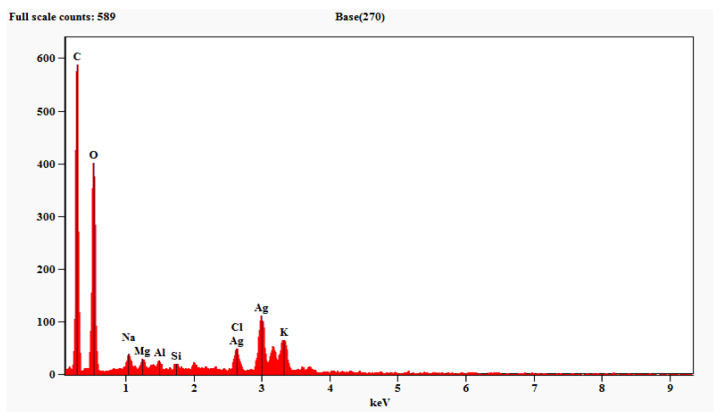
EDS spectra of silver nanoparticles synthesized using *Trichoderma harzianum* filtrate.

**Figure 4 biomolecules-11-00535-f004:**
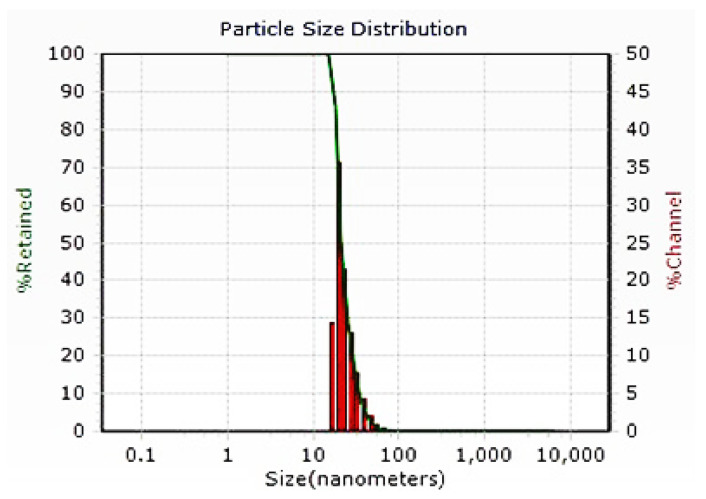
DLS analysis of silver nanoparticles synthesized using *Trichoderma harzianum* filtrate.

**Figure 5 biomolecules-11-00535-f005:**
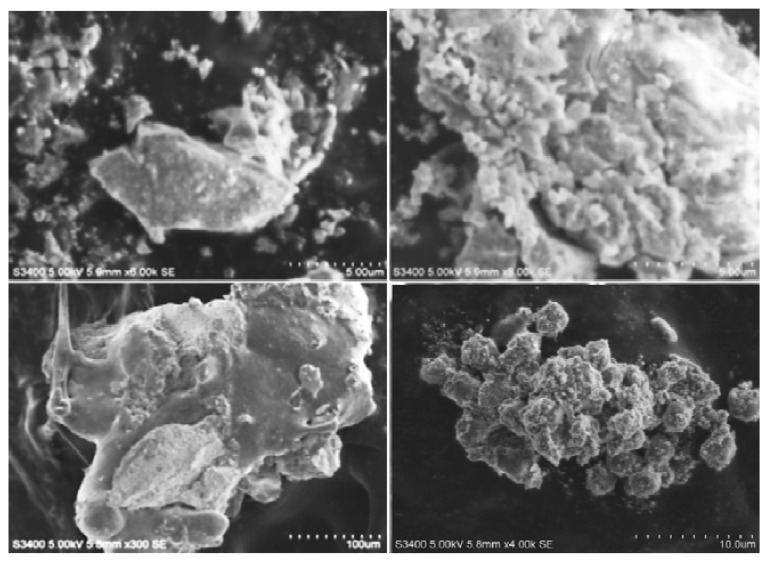
Scanning electron microscopy analysis of silver nanoparticles synthesized using *Trichoderma harzianum* filtrate.

**Figure 6 biomolecules-11-00535-f006:**
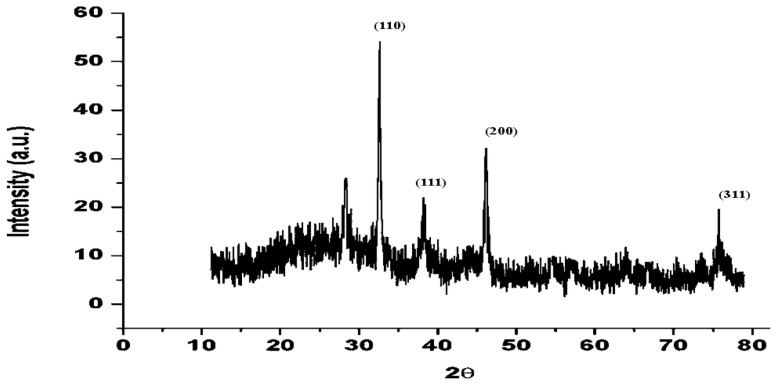
XRD analysis of silver nanoparticles synthesized using *Trichoderma harzianum* filtrate.

**Figure 7 biomolecules-11-00535-f007:**
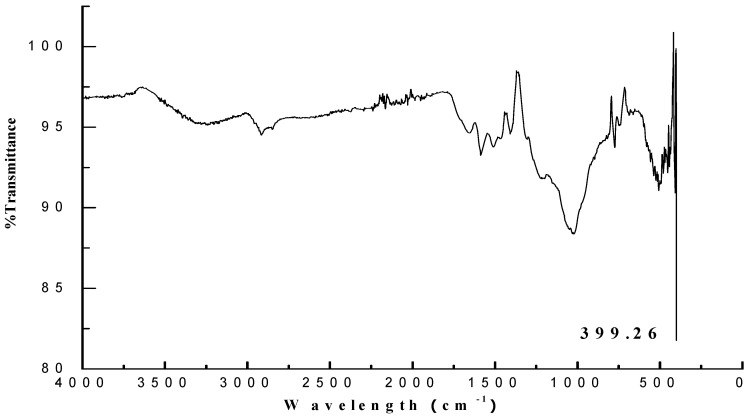
FT-IR spectra of silver nanoparticles synthesized from *Trichoderma harzianum* filtrate.

**Figure 8 biomolecules-11-00535-f008:**
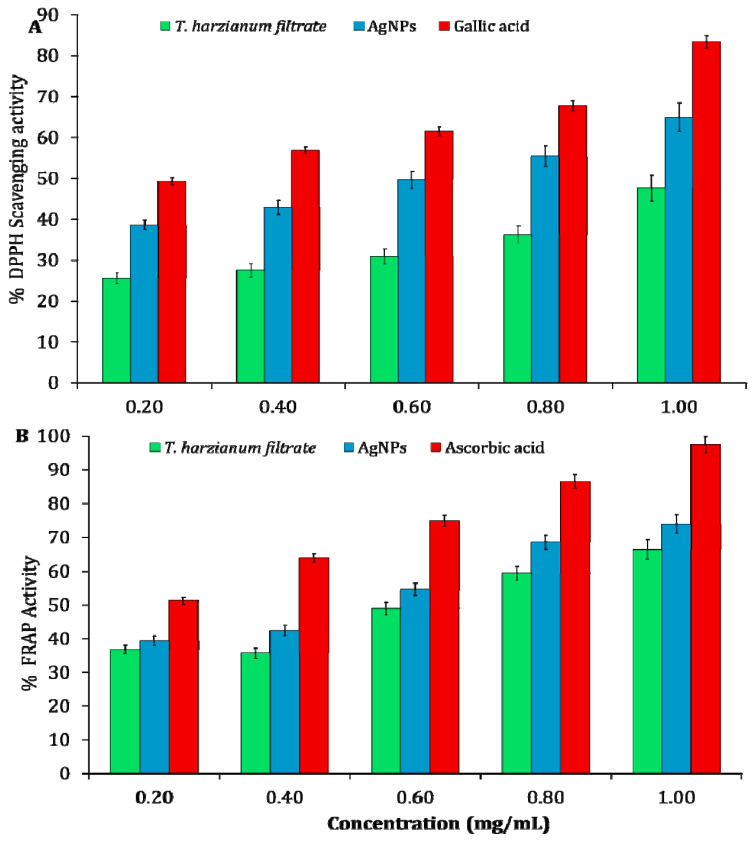
(**A**) Estimation of DPPH radical scavenging activity and (**B**) ferric reducing antioxidant power activity from different concentrations of silver nanoparticles synthesized using *Trichoderma harzianum* filtrate.

**Figure 9 biomolecules-11-00535-f009:**
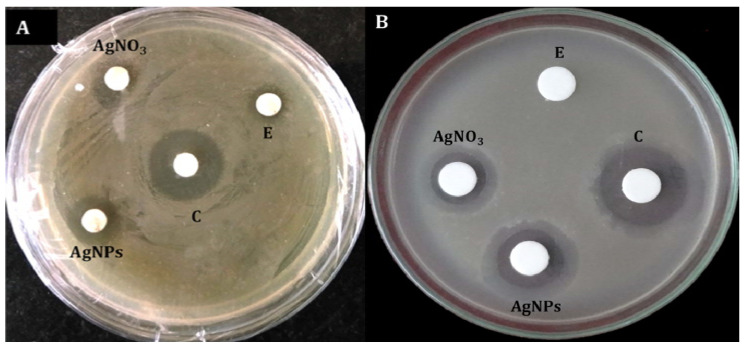
Antibacterial activity of silver nanoparticles synthesized using *Trichoderma harzianum* filtrate, as determined by disc diffusion method. (**A**) *Staphylococcus aureus* (Gram-positive); (**B**) *Ralstonia solanacearum* (Gram-negative). C: Positive control (streptomycin)*;* AgNPs: silver nanoparticles; AgNO_3_: silver nitrate; E: negative control (*Trichoderma harzianum* culture filtrate).

**Figure 10 biomolecules-11-00535-f010:**
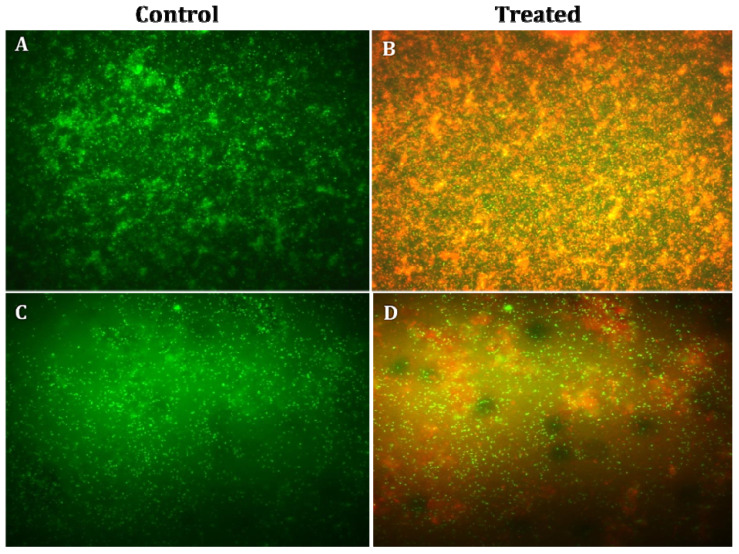
Fluorescence microscopy images of control and AgNP-treated *Staphylococcus aureus* and *Ralstonia solanacearum*: (**A**) control *S. aureus*; (**B**) *S. aureus* treated with silver nanoparticles; (**C**) control *R. solanacearum*; (**D**) *R. solanacearum* treated with silver nanoparticles. Green spots represent live bacterial cells, whereas red fluorescence indicates dead bacteria.

**Figure 11 biomolecules-11-00535-f011:**
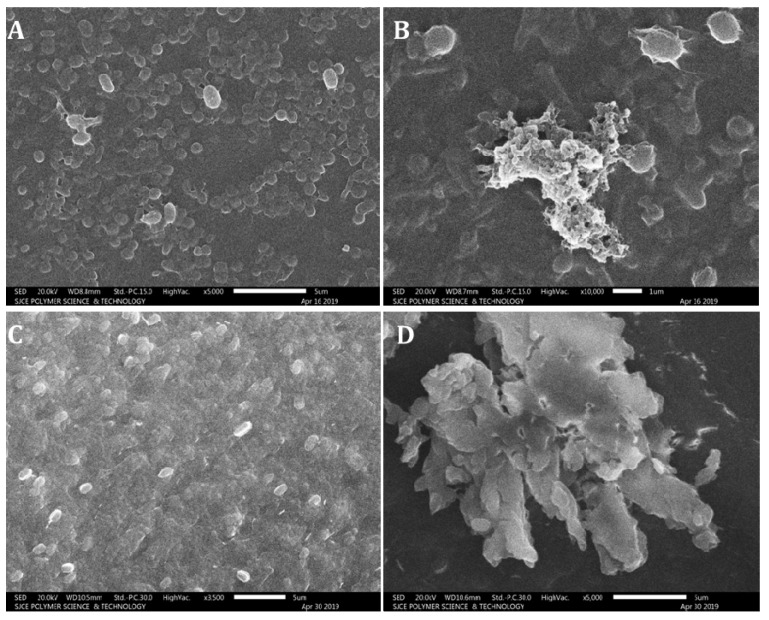
Scanning electron microscopy images of control and AgNP-treated *Staphylococcus aureus* and *Ralstonia solanacearum*: (**A**) control *S. aureus*; (**B**) *S. aureus* treated with silver nanoparticles; (**C**) control *R. solanacearum*; (**D**) *R. solanacearum* treated with silver nanoparticles. Silver nanoparticles cause morphological modification of bacterial cell structures.

**Figure 12 biomolecules-11-00535-f012:**
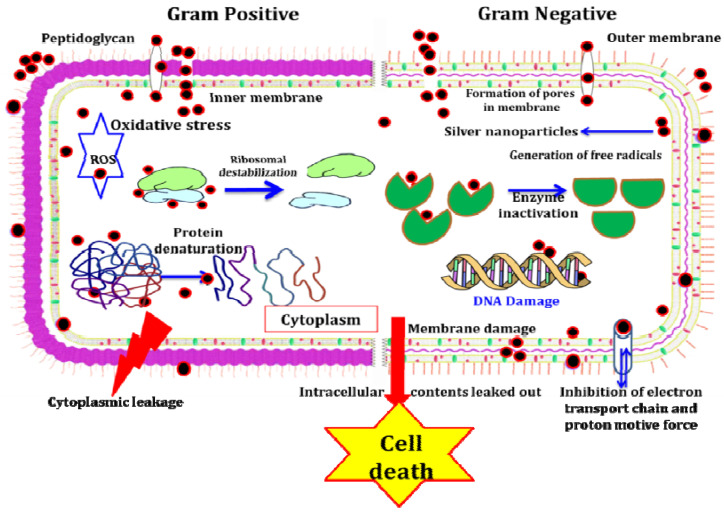
Proposed various modes of action of silver nanoparticles against bacterial growth/proliferation.

**Table 1 biomolecules-11-00535-t001:** Antibacterial compounds identified in *Trichoderma harzianum* filtrate using LC-MS/MS.

Sl. No.	*m*/*z* Obtained	Actual Mass	Error	Molecular Formula	Tentative Identification	Biological Activity	References
1.	489.2323	489.2323	0.0	C_28_H_33_N_4_O_2_S	1-Benzoyl-3-[(*S*)-((2*S*,4*R*,8*R*)-8-ethylquinuclidin-2-yl](6-methoxyquinolin-4-yl)methyl)thiourea	Antibacterial activities	[29]
2.	416.2064	416.382	0.1756	C_21_H_20_O_9_	Puerarin	Antimicrobial and antioxidant activities	[30,31]
3	432.2086	432.37	0.1614	C_21_H_20_O_10_	Genistein	Antimicrobial and antioxidant activities	[32]
4	407.2975	407.5	0.2025	C_23_H_37_NO_5_	Isotalatizidine	Antibacterial activities	[33]
5.	800.5387	801.01	0.4713	C_42_H_72_O_14_	Ginsenoside	Antimicrobial activities	[34]

**Table 2 biomolecules-11-00535-t002:** Percentage by weight of metallic elements present in silver nanoparticles from *T. harzianum* filtrate.

Element Line	Weight (%)	Weight % (Error)	Atom (%)
C K	0.00	---	0.00
O K	0.00	---	0.00
Na K	12.34	±1.20	27.73
Mg K	3.91	±0.72	8.32
Al K	3.01	±0.57	5.77
Cl K	7.54	±1.07	10.99
Cl L	---	---	---
K K	14.43	±0.79	19.06
K L	---	---	---
Ag L	58.75	±4.45	28.13
Ag M	---	---	---
Total	100.00		100.00

**Table 3 biomolecules-11-00535-t003:** Zone of inhibition of antibacterial activity exhibited by silver nanoparticles (AgNPs) from *T. harzianum.*

Microorganisms	Zone of Inhibition (mm)			
	AgNPs	Silver Nitrate	*T. harzianum*filtrate	Streptomycin (25 μg/disc)
Antibacterial activity				
*Staphylococcus aureus*	14.6 ± 2.33 ^bc^	2.3 ± 0.23 ^a^	4.3 ± 0.66 ^c^	16.89 ± 0.54 ^a^
*Bacillus subtilis*	13.86 ± 0.57 ^a^	2.9 ± 0.56 ^c^	4.9 ± 0.66 ^d^	25.56 ± 0.78 ^c^
*Escherichia coli*	15.56 ± 1.67 ^d^	2.7 ± 0.54 ^b^	2.9 ± 0.54 ^b^	25.33 ± 0.6 ^b^
*Ralstonia solanacearum*	17.43 ± 1.23 ^e^	4.56 ± 0.89 ^de^	1.0 ± 0.08 ^a^	20.21 ± 0.48 ^d^

The data represent mean ± SE of replicates (*n* = 3). Data with different superscript letters are significantly different between treatments and control, as determined by Duncan’s multiple range test, at *p* ≤ 0.05.

**Table 4 biomolecules-11-00535-t004:** Minimum inhibitory concentration of AgNPs from *Trichoderma harzianum* against bacterial pathogens.

Pathogens	4096	2048	1024	512	256	128	64	32	16	8	MIC (μg mL^−1^)
*S. aureus*	-	-	-	-	-	+	+	+	+	+	256
*B. subtilis*	-	-	-	-	+	+	+	+	+	+	512
*E. coli*	-	-	-	-	-	-	+	+	+	+	128
*R. solanacearum*	-	-	-	-	-	-	-	+	+	+	64
Streptomycin (positive control)	-	-	-	-	-	-	-	-	-	+	16
Culture filtrate (negative control)	-	-	-	+	+	+	+	+	+	+	1024

Note: - represents no growth of bacterial pathogens; + represents the growth of bacterial pathogens; MIC: minimum inhibitory concentration.

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
