# Peer review of "Ameliorated Antibacterial and Antioxidant Properties by Trichoderma harzianum Mediated Green Synthesis of Silver Nanoparticles"

_biomolecules, 2021, doi:10.3390/biom11040535_

Round 1

Reviewer 1 Report

In the article: "Ameliorated antibacterial and antioxidant properties by Trichoderma harzianum mediated green synthesis of silver nanoparticles," the Authors investigated the fungus Trichoderma harzianum use in the synthesis of extracellular AgNPs, and detected different bioactive metabolites in fungus cell filtrate, based on LC-MS/MS analysis. In my opinion, the paper presents interesting results. Below I present some suggestions to improve the manuscript.

Methods:

  • Paragraph 2.5. - did the blank contain some medium in place of AgNPs? The volume of the blank should be the same as the sample. Please add this information. Moreover, in the methodology, the Authors should add in what way the standard was prepared. The name of reference substances, concentrations, and type of medium use to dissolve them is necessary for the methodology.
  • Paragraph 2.5.2. - the reference of literature is needed for the used method. How were the results expressed? Authors should add this information in this place. Authors should add in this section in what way the standard was prepared.
  • It could also be interesting to compare the antioxidant activities of cell filtrate and AgNPs nanoparticles. I do not see that comparison in this manuscript.

Results:

  • Paragraphs 3.3.1 – the concentrations indicated are the concentrations prepared to study or the concentration in the sample? .Authors should add the more precise information
  • Paragraph 3.3.2. – the same comment as above
  • FRAP describe "Ferric Reducing Antioxidant Power," it is not scavenging activity
  • Table 3: - R. solanacearum should be as Ralstonia solanacearum, or all the names should be: S. aureus, B. subtilis, etc. then explained below the Table
  • Figure 9: what means "control" - which substance? It could be defined in the caption
  • Table 4: the Subcaption are carelessly prepared
  • On the chromatograms/spectra, the name/structure of identified compounds should be included (Figure 1)
  • In the captions, Authors could note the full name of Trichoderma
  • Figure 4 should be in the better quality

Discussion:

  • Can the Authors explained to me in what way they calculated IC50 for FRAP analysis? This method, following my knowledge, runs differently from DPPH analysis. How had the Authors designated % of sample activity?
  • The discussion concerning paragraphs on antimicrobial and antioxidant activities could present some information connecting the chemical compound's role in the cell filtrate.

Others:

  • There are minor punctuation errors in the work, such as two dots instead of one (line 70), two or no spaces (in many places).
  • On lines 89 and 98, and 103, in my opinion, the full name Trichoderma harzianum is unnecessarily used. In my opinion, T. harzianum would be sufficient.
  • Sometimes, the bottom index is missing (line 136)
  • Chromatograms/spectra are sometimes too extended or too narrowed
  • The subtitles of the Figures and Tables should be deprived of editorial errors.

Author Response

Responses to the Reviewers’ Comments

As per recommendations of reviewers, the article has been revised, re-written wherever required and suggested corrections have been incorporated accordingly. We hope the article is in order and fulfills the comments raised.

Reviewer #1:

In the article: "Ameliorated antibacterial and antioxidant properties by Trichoderma harzianum mediated green synthesis of silver nanoparticles," the Authors investigated the fungus Trichoderma harzianum use in the synthesis of extracellular AgNPs, and detected different bioactive metabolites in fungus cell filtrate, based on LC-MS/MS analysis. In my opinion, the paper presents interesting results. Below I present some suggestions to improve the manuscript.

Response: We are very glad that the Reviewer highly evaluated our manuscript, and provided constructive comments and suggestions that have helped us improve the quality of our manuscript.

Methods:

Comment 1: Paragraph 2.5. did the blank contain some medium in place of AgNPs? The volume of the blank should be the same as the sample. Please add this information. Moreover, in the methodology, the Authors should add in what way the standard was prepared. The name of reference substances, concentrations, and type of medium use to dissolve them is necessary for the methodology.

Response: We would like to express our special thanks to Reviewer for critical observation of our manuscript. We have now incorporated the detail methodology, reference substances, and concentrations in the revised manuscript.

The biosynthesized AgNPs and the culture filtrate were used to assess the antioxidant property by DPPH radical scavenging assay [25]. 1.5 mL of freshly prepared DPPH (4 mg of DPPH in 100 mL of 95% ethyl alcohol) was added to 1.5 mL of culture filtrate and AgNPs samples (0.20 - 1.0 mg/mL) respectively; incubated at room temperature in dark for 30 min and reduction of DPPH was determined spectrophotometrically at 517 nm against blank (1.5 mL of DPPH solution and 1 mL of 95% ethanol) and gallic acid was used as standard (0.2-1.0 mg/mL in 95% ethyl alcohol). The blank consisted of 1.5 mL of DPPH solution containing the filtrate. The experiments were repeated thrice. The percent activity and IC50 (concentration of sample needed to inhibit 50% DPPH) was determined (L.173-183). 

Comment 2: Paragraph 2.5.2. - The reference of literature is needed for the used method. How were the results expressed? Authors should add this information in this place. Authors should add in this section in what way the standard was prepared.

Response: We highly appreciate the Reviewer for this suggestion. Accordingly, we have included the below appropriate reference [26] in connection with the methods and preparation of standard in the revised manuscript.

“[26] Pulido, R.; Bravo, L.; Sauro-Calixto, F. Antioxidant activity of dietary polyphenols as determined by a modified ferric reducing/antioxidant power assay. J. Agric. Food Chem. 2000, 48, 3396-3402”.

The antioxidant potential of AgNPs and the culture filtrate was analysed by Ferric-Reducing Antioxidant Power (FRAP) assay [26]. The FRAP reagent (4.5 mL) was prepared by mixing 2.5 mL of TPTZ (2, 4, 6-tripyridyl-striazine) solution (10 mM TPTZ in 40 mM HCl) and 20 mM FeCl3 in 25 mL of acetate buffer (0.3 M, pH 3.6), mixed with 0.5 mL of test samples at different concentrations (0.2-1.0 mg/mL). Deionized water and ethanol was served as blank. The reaction mixture was incubated at 37 °C for 30 min and the absorbance was recorded at 593 nm. The dark blue color formed as Fe3+–TPTZ complex was reduced to Fe2+–TPTZ. Freshly prepared aqueous ascorbic acid solution (0.2-1.0 mg/mL) was used as standard (L.188-196).

Comment 3: It could also be interesting to compare the antioxidant activities of cell filtrate and AgNPs nanoparticles. I do not see that comparison in this manuscript.

Response: Thank you so much for this comment and suggestion. The aim of the experiment is to determine the antioxidant activity in both the test samples (AgNPs and the culture filtrate) which we have successfully carried out in this study. The results of our study indicated that AgNPs exhibited maximum DPPH scavenging activity as compared with fungal filtrate. The comparative efficacy of antioxidant activity in both the test samples was discussed in the discussion section (L.337-339).

Results:

Comment 4: Paragraphs 3.3.1 – the concentrations indicated are the concentrations prepared to study or the concentration in the sample? Authors should add the more precise information

Response: We thank the reviewer for his/her critical observation and providing the valuable suggestion. The AgNPs were synthesized from culture filtrate using various concentrations of 0.2, 0.4, 0.6, 0.8, and 1.0 mg/mL respectively for DPPH scavenging activity (L.332-333).

Comment 5: Paragraph 3.3.2. – the same comment as above

Response: Thank you so much for this comment which indeed connected to the above comment No. 4. The AgNPs were synthesized from culture filtrate. The AgNPs were used at concentrations of 0.2, 0.4, 0.6, 0.8, and 1.0 mg/mL respectively for FRAP assay (L. 344).

Comment 6: FRAP describe "Ferric Reducing Antioxidant Power," it is scavenging activity?

Response: We thank the reviewer for this critical observation. Sorry for this topographical error, DPPH refers to scavenging activity. Appropriate correction has been made in the revised manuscript (L.344).

Comment 7: Table 3: - R. solanacearum should be as Ralstonia solanacearum, or all the names should be: S. aureus, B. subtilis, etc. then explained below the Table

Response: As recommended, the names of the organisms are now re-written with full scientific names in Table 3 in the revised manuscript (Page No. 15).

Comment 8: Figure 9: what means "control" - which substance? It could be defined in the caption

Response: Thank you so much for this valuable comment and enquiry. We have used Streptomycin (25μg/disc) as positive control and the same is now incorporated in the revised manuscript (Page No. 15).

Comment 9: Table 4: the Subcaption are carelessly prepared

Response: We are extremely sorry for this mistake. To avoid confusion, we have now revised the subcaption as (-) represents the no growth of bacterial pathogens; (+) represents the growth of bacterial pathogens; MIC- Minimum Inhibitory Concentration in the revised manuscript (L.381).

Comment 10: On the chromatograms/spectra, the name/structure of identified compounds should be included (Figure 1)

Response: Thank you so much for this comment and suggestion. Accordingly, the name/structure of identified compounds are included in Figure 1 of the revised manuscript (Page No. 7 & 8).

Comment 10: In the captions, Authors could note the full name of Trichoderma

Response: We appreciate the reviewer for this comment and suggestion. To meet the Reviewer suggestion, we have incorporated the genus and species name “Trichoderma harzianum” in the revised manuscript (L.288, 301, 307, 320, 329, 353, 369).

Comment 11: Figure 4 should be in the better quality

Response: Thank you so much for this comment. We have now provided a new Figure 4with high resolution (Page No. 11).

Discussion:

Comment 12: Can the Authors explained to me in what way they calculated IC50 for FRAP analysis? This method, following my knowledge, runs differently from DPPH analysis. How had the Authors designated % of sample activity?

Response: Thank you so much for this comment. The detail methodology of antioxidant activities (DPPH and FRAP) has been incorporated in the revised manuscript. The IC50 value was calculated by plotting the inhibitor concentration against percent activity ([I] Vs Activity %). The IC50 value was determined by the linear (y=mx+n) or parabolic (y=ax2+bx+c) equation on graph, at y=50, x value was considered as IC50 value.

Comment 13: The discussion concerning paragraphs on antimicrobial and antioxidant activities could present some information connecting the chemical compound's role in the cell filtrate.

Response: We thank the reviewer for this important comment. As suggested the following text describing the role of compounds responsive for antimicrobial and antioxidant properties has been included in the revised manuscript.

“The Trichoderma species stimulates the production of low-molecular weight and volatile compounds like phytoalexins, harzianopyridone, pyrones, terpenes, peptaibols, terpenoid compounds that are reported previously to have antibacterial activity [57]. Some Trichoderma species also have been known to produce significant secondary metabolites like plant growth regulators, antibiotics and enzymes which protect plants from infecting pathogens [58]. The enzymes secreted by the Trichoderma species are reported to exhibit antimicrobial, anticancer and antioxidant activity [59]” (L.503-509).  

Others:

Comment 14: There are minor punctuation errors in the work, such as two dots instead of one (line 70), two or no spaces (in many places).

Response: We are extremely sorry for this error/s. All these punctuation errors have been fixed properly in the entire revised manuscript.

Comment 15: On lines 89 and 98, and 103, in my opinion, the full name Trichoderma harzianum is unnecessarily used. In my opinion, T. harzianum would be sufficient.

Response: We thank the reviewer for this observation. We have taken care to mention the name “Trichoderma harzianum” at its first usage, followed by abbreviation (T. harzianum) in each section of the manuscript (L.104, 106).

Comment 16: Sometimes, the bottom index is missing (line 136)

Response: Thank you so much for pointing this. We have now fixed the bottom index for easy visibility in the revised manuscript (L.120-135).

Comment 17: Chromatograms/spectra are sometimes too extended or too narrowed

Response: We highly appreciate the reviewer for his/her critical observation of our manuscript. We have now supplied the spectra with uniform in size

Comment 18: The subtitles of the Figures and Tables should be deprived of editorial errors.

Response: Thank you so much again for this comment and suggestion. The subtitles of Figures and Tables are now fixed without error and easy to understand in the revised manuscript.

Reviewer 2 Report

The authors describe the green synthesis of silver nanoparticles (NP) using the culture filtrate of the biocontrol agent fungus Trichoderma harzianum. The authors characterise the culture filtrate by LC-MS/MS and identify five potential components within that have previously been shown to have biological properties, which could be attributed to the biological effect of the resulting NP. The synthesised NP were physically characterised extensively using a variety of methods. The authors then assess the NPs with regards to their biological activity, such as radical scavenging and growth inhibition, showing a generally positive effect, suggesting suitability of these NPs as biocontrol agents. This is an interesting study with intriguing findings, highlighting how naturally occurring biocontrol agents could be harnessed in non-traditional ways to synthesise materials with antimicrobial properties.

Some comments regarding reader experience:

1) The language quality deteriorates through the manuscript, such that some sentences are incoherent towards the end. The manuscript would greatly benefit from editing by an English-speaking person, possibly the same person that has written/edited the early parts of the manuscript.

2) There are several typo’s, missing spaces, etc that should be fixed.

I recommend the following improvement:

The authors should include the following controls in all biological experiments: NPs that were synthesised in the absence of culture filtrate, and culture filtrate alone. This would provide insight into the contribution of the culture filtrate to the observed properties of the synthesised NPs. The authors state that silver itself has been shown to have beneficial medical properties, so using a control that would quantify additional benefits derived from the culture media would be immensely informative and elevate this manuscript. Further, although possible for some experiments, no statistical analysis was performed to assess whether the observed inhibitory effect is significant compared to controls, this should be included and would strengthen the results.

The characterisation of the filtrate identified 5 components with known biological activity. However, the spectra presented also show two other abundant peaks at 1218.8970 and 1219.9059 m/z, could the authors comment on the identify of those?

Figure 2 only shows the UV spectrum at the end of the synthesis, but why not show both spectra to illustrate the change due to NP formation, given that the authors state that they took both measurements this would be useful?

Using dynamic light scattering identified the particles to be of: “23.52nm in diameter and 14.64nm in width”. What is the width here, are these particles not spherical?

Figure 5 shows NPs of seemingly very different sizes. Specifically, the image presented on the bottom right compared to the image on the top right, even though the scale is roughly similar. Could the authors clarify this, maybe highlight a particle?

The authors draw attention to 4 peaks in the XRD analysis (L304-308) and then state that 3 other peaks are likely due to impurities. Could the authors add an arrow to the Figure to show these peaks, as I can only see one of the 3 (at 27), but not the other two. If the other two peaks are so elusive, then why are they recognised as distinct peaks, please clarify this?

Figure 8 shows the DPPH and FRAP activity of the NPs from which the authors calculate the IC50. The authors should show the correlation curves used for this calculation. Further, the FRAP IC50 given is not supported by the figure, which instead suggests that the concentrations tested in fact never resulted in 50% FRAP activity.

The representative images shown in Figure 9 do not support the measures given in Table 3. For example, the R. solanacearum images show no growth inhibition by the filtrate or the AgNO3, yet the table states a clear inhibition. Further, in the presented images the streptomycin zone of inhibition of R. solanacearum is very much larger than for S. aureus, yet the table does not reflect this. There are further inconsistencies that need to be addressed and corrected. Additionally, the authors should add a statement towards the expected vs observed outcome for the positive control streptomycin.

In Table 4 the authors summarise the MIC of the NPs against the pathogens, but the numbers do not fit with the reported observation of growth. I suspect S. aureus and E. coli have been mixed up, but the MIC for R. solanacearum is wrong. Also, the authors should include a positive and negative control as well as replication.

Could the authors address the following minor points, to clarify the manuscript:

As a general comment, volume statements are not insightful, instead the concentrations should be stated. Could the authors adjust all microscopic images such that the scale bar is identical across image sets, this will facilitate comparisons.

Section 2.2:

L116: The authors state that 2ug/ml culture filtrate was used for LC-MS/MS. Was the culture filtrate completely dried and then resuspended to achieve that concentration? If so, I’m curious to know how much dried filtrate was obtained.

L123: The authors state that 0.5g of sample was diluted prior to LC-MS/MS analysis, how does that reconcile with the 2ug/ml concentration stated above? And how does this work with the 5ul volume that was injected for analysis, which seems very little material if the concentration was 2ug/ml but quite a lot if 0.5g were used? I think there is some confusion about volumes/concentrations/masses that need to be clarified.

Section 2.3:

L135: The culture filtrated used for NP synthesis, was that the 2ug/ml solution also used for LC-MS/MS or is this a different concentration?

L136-137: Was the temperature 25C or room temperature?

Section 2.5.2:

L191-192: Why does the blank contain a standard?

Section 2.6.2:

The authors should state the negative and positive controls used in this experiment?

Section 2.7:

The authors need to specify the positive and negative controls used and add information about replication of this experiment. From the results, I gather this was only performed once. Given that it is a comparatively simple experiment, the authors should perform it in triplicates, which would also allow statistical analysis.

Section 3.3.1:

L322-323: It is unclear what the concentration range refers to. The results suggest that this was the concentration range of the synthesised NP used for this experiment. Please modify to make this clearer.

Section 3.3.2:

L332-333: “The results of this study indicated that biosynthesised AgNPs exhibited maximum FRAP activity.” What do the authors mean here? The range is given as 22-47%, I would understand 100% as maximum activity, which is not reached. Please amend to clarify this.

Section 4:

L408 & 414: The authors refer to “earlier reports”, please include citations.

L422-423: The order in which inhibition is reported is not correct, the most inhibited microbe is R. solanacearum with the least being B. subtilis. This should be rectified or reworded.

L432 onwards: The authors at length discuss the previously reported properties of AgNP. For the most part of this paragraph, it is unclear whether in those NP were green synthesised or not. This needs to be made very clear, as the relevance and comparison with the presented study and the discussion is critically based on this distinction.  

L444: “captivates” should be changed to “captures”

L455-459: These sentences are incoherent. The manuscripts language quality deteriorates significantly towards the end, which should be addressed.

L493: The legend is incoherent.

Section 5:

L496: “significant thrust” should be changed to “significantly growing”.

L504: What are the other biological activities the authors are referring to?

Author Response

Reviewer 2:

Response: We would like to express our special thanks to this Reviewer for positively evaluating our manuscript and also providing constructive comments which will help us to improve the quality of our manuscript.

Comment 1: The language quality deteriorates through the manuscript, such that some sentences are incoherent towards the end. The manuscript would greatly benefit from editing by an English-speaking person, possibly the same person that has written/edited the early parts of the manuscript.

Response: Thank you so much for this comment and suggestion. The manuscript is now edited by Dr. Savitha De Britto who is an Native English Speaker and one of the co-author in this manuscript.

Comment 2: There are several typo’s, missing spaces, etc that should be fixed.

Response: We are extremely sorry for this topographical error/s. All these errors including spaces are now fixed in the revised manuscript.

I recommend the following improvement:

The authors should include the following controls in all biological experiments: NPs that were synthesised in the absence of culture filtrate, and culture filtrate alone. This would provide insight into the contribution of the culture filtrate to the observed properties of the synthesised NPs. The authors state that silver itself has been shown to have beneficial medical properties, so using a control that would quantify additional benefits derived from the culture media would be immensely informative and elevate this manuscript. Further, although possible for some experiments, no statistical analysis was performed to assess whether the observed inhibitory effect is significant compared to controls, this should be included and would strengthen the results.

Response: Thank you so much for this comment and suggestion. In the present study NPs were synthesized from cell filtrates. Since cell filtrate was used along with Ag for synthesize of AgNPs, culture filtrate alone has been used as control. Throughout the experiment we have maintained appropriate controls and the replicated data were presented after SPSS statistical analysis.

Comment 3: The characterization of the filtrate identified 5 components with known biological activity. However, the spectra presented also show two other abundant peaks at 1218.8970 and 1219.9059 m/z, could the authors comment on the identify of those?

Response: Thank you so much for this important observation and comment. In the present research work only antimicrobial compounds are focused. The two other abundant peaks at 1218.8970 and 1219.9059 m/z do not represent any significant antimicrobial compounds from the literature reviewed and hence not included.

Comment 4: Figure 2 only shows the UV spectrum at the end of the synthesis, but why not show both spectra to illustrate the change due to NP formation, given that the authors state that they took both measurements this would be useful?

Response: We thank the Reviewer again for this comment and inquiry. The aim of the present work is to synthesis of NPs from Microbial source. Hence, the change in the color was measured after synthesis by UV spectrum.

Comment 5: Using dynamic light scattering identified the particles to be of: “23.52 nm in diameter and 14.64 nm in width”. What is the width here, are these particles not spherical?

Response: We would like to express our special thanks to Reviewer for this query. The size of biosynthesized AgNPs was recorded within a range of 14.64 to 23.52 nm in diameter. The average size of the AgNPs was 21.49nm. The SEM image has been replaced with 10µm magnification and AgNPs appears to be spherical in shape (Page No.11).

Comment 6: Figure 5 shows NPs of seemingly very different sizes. Specifically, the image presented on the bottom right compared to the image on the top right, even though the scale is roughly similar. Could the authors clarify this, maybe highlight a particle?

Response: The authors thank this Reviewer for this critical observation, comment and suggestion. Following your advice, corrections have been incorporated in the revised manuscript. The magnification of these images in Figure 5 was different and hence, the AgNPs appears to be different in size. The AgNPs are spherical in shape but, the secondary metabolites present in the fungal extract adhere to AgNPs and make them to appear in irregular shapes (Page No.11).

Comment 7: The authors draw attention to 4 peaks in the XRD analysis (L304-308) and then state that 3 other peaks are likely due to impurities. Could the authors add an arrow to the Figure to show these peaks, as I can only see one of the 3 (at 27), but not the other two. If the other two peaks are so elusive, then why are they recognized as distinct peaks, please clarify this? Furthermore, no additional peaks in the XRD were observed for Ag2O revealing the high purity of the as synthesized silver crystal.

Response: We apologize for mentioning two extra peaks in XRD. The experiment was carried out in three different trials and these peaks were obtained only in one of the trials; the authors included only the significant peaks in the revised manuscript. 

“The XRD patterns showed peaks of AgNPs at 32.3, 38, 46 and 77 corresponding to (110), (111), (200) and (311) planes of AgNPs. The XRD study confirmed that the particles were AgNPs with face centered, cubic crystal structure. The peak in diffraction pattern at 27 degree could be due to bioorganic impurities” (L. 311-316).

Comment 8: Figure 8 shows the DPPH and FRAP activity of the NPs from which the authors calculate the IC50. The authors should show the correlation curves used for this calculation. Further, the FRAP IC50 given is not supported by the figure, which instead suggests that the concentrations tested in fact never resulted in 50% FRAP activity.

Response: We highly appreciate the Reviewer for this comment. The detail methodology of antioxidant activities (DPPH and FRAP) has been incorporated in the revised manuscript (Section 2.5.1. and 2.5.2).

“The IC50 value was calculated by plotting the inhibitor concentration against percent activity ([I] Vs Activity %). The IC50 value was determined by the linear (y=mx+n) or parabolic (y=ax2+bx+c) equation on graph, at y=50, x value was considered as IC50 value”.

Comment 9: The representative images shown in Figure 9 do not support the measures given in Table 3. For example, the R. solanacearum images show no growth inhibition by the filtrate or the AgNO3, yet the table states a clear inhibition. Further, in the presented images the streptomycin zone of inhibition of R. solanacearum is very much larger than for S. aureus, yet the table does not reflect this. There are further inconsistencies that need to be addressed and corrected. Additionally, the authors should add a statement towards the expected vs observed outcome for the positive control streptomycin.

Response: We would like to express our special thanks to Reviewer for this suggestion. We are sorry for this wrong interpretation of the data. The antibacterial activity of AgNPs (Figure 9 and Table 3) has corrected in the revised manuscript (Page No. 15).

Comment 10: In Table 4 the authors summarise the MIC of the NPs against the pathogens, but the numbers do not fit with the reported observation of growth. I suspect S. aureus and E. coli have been mixed up, but the MIC for R. solanacearum is wrong 64. Also, the authors should include a positive and negative control as well as replication.

 Response: Yes, we totally agree with your comment and suggestion. The antibacterial activity of AgNPs and MIC for R. solanacearum from Table 4 is revised. The suggested corrections are incorporated in the revised manuscript and data is correctly represented in table 4 with negative and positive controls. 

Section 2.2:

Comment 11: L116: The authors state that 2 ug/ml culture filtrate was used for LC-MS/MS. Was the culture filtrate completely dried and then resuspended to achieve that concentration? If so, I’m curious to know how much dried filtrate was obtained.

L123: The authors state that 0.5g of sample was diluted prior to LC-MS/MS analysis, how does that reconcile with the 2ug/ml concentration stated above? And how does this work with the 5 ul volume that was injected for analysis, which seems very little material if the concentration was 2 ug/ml but quite a lot if 0.5g were used? I think there is some confusion about volumes/concentrations/masses that need to be clarified.

Response: The authors thank reviewer for his/her critical observation and comment. To avoid confusion, section 2.2 is completely revised as follows:

“The chemical constituents from culture filtrate were determined using LC-MS/MS. The 50 mg of T. harzianum culture filtrate extract was suspended in 2 mL of methanol and filtered through 0.22 µm nylon membrane prior to injection. HPLC was interfaced with a Q-TOF mass spectrometer fitted with an ESI source. HPLC column Phenomenex 5 μ C8, (150 × 2 mm id.) was used for the analysis. The solvents were delivered at a total flow rate of 0.1 mL/min and run by isocratic elution. The MS spectra were acquired in the positive ion mode. The temperature of the drying gas (N2) was 350 °C, at a gas flow rate of 6 mL/min, and a nebulizing pressure (N2) of 25 psi. A 20 μL volume of fungal extract was injected onto the analytical column for analysis. The mass fragmentations were identified by using spectrum database for organic compounds. The analytical LC/MS experiment was performed in using TSQ Quantum Access MAX Triple- Stage Quadrupole Mass Spectrometer. Waters Mass Lynx and Target Lynx softwares were used for data acquisition and data processing respectively. The MS analysis was performed using ESI in the positive mode. The MS parameters were curtain gas 10, gas1 20 and gas 20, needle voltage 5000 V and declustering potential 100 V. TOF was operated between 50 and 1500 m/z with low mass resolution of 4.7 and high mass resolution of 15” (L120-135).

Section 2.3:

Comment 12: L135: The culture filtrated used for NP synthesis, was that the 2ug/ml solution also used for LC-MS/MS or is this different concentration?

Response Thank you so much for this comment and suggestion which is indeed connected to above comment. A 20 μL volume of the fungal extract was injected onto the analytical column for analysis.  Hence, we have rewritten the paragraph in the revised manuscript (L120-135).

Comment 13: L136-137: Was the temperature 25 ºC or room temperature?

Response: We would like to express our special thanks to Reviewer for this enqury. It is 25 ºC and has been incorporated in the revised manuscript (L139).

Section 2.5.2:

Comment 14: L191-192: Why does the blank contain a standard?

Response: The Section 2.5.2 (Ferric reducing antioxidant power) method has been rewritten in the revised manuscript as follows:

“The antioxidant potential of AgNPs and the culture filtrate was analysed by Ferric-Reducing Antioxidant Power (FRAP) assay [26]. The FRAP reagent (4.5 mL) was prepared by mixing 2.5 mL of TPTZ (2, 4, 6-tripyridyl-striazine) solution (10 mM TPTZ in 40 mM HCl) and 20 mM FeCl3 in 25 mL of acetate buffer (0.3 M, pH 3.6), mixed with 0.5 mL of test samples at different concentrations (0.2-1.0 mg/mL). Deionized water and ethanol was served as blank. The reaction mixture was incubated at 37 °C for 30 min and the absorbance was recorded at 593 nm. The dark blue color formed as Fe3+–TPTZ complex was reduced to Fe2+–TPTZ. Freshly prepared aqueous ascorbic acid solution (0.2-1.0 mg/mL) was used as standard” (L188-196).

Section 2.6.2:

Comment 15: The authors should state the negative and positive controls used in this experiment?

Response: We have used Streptomycin as positive control at a concentration of 25µg/mL, fungal filtrate and AgNO3 for the comparison of AgNPs activity (L.211).

Section 2.7:

Comment 16: The authors need to specify the positive and negative controls used and add information about replication of this experiment. From the results, I gather this was only performed once. Given that it is a comparatively simple experiment, the authors should perform it in triplicates, which would also allow statistical analysis.

Response: The experiments were repeated thrice and mean values were recorded and incorporated in the revised manuscript (L. 216-226).

Section 3.3.1:

Comment 17: L322-323: It is unclear what the concentration range refers to. The results suggest that this was the concentration range of the synthesised NP used for this experiment. Please modify to make this clearer.

Response: Thank you so much for this comment. The concentration range of the biosynthesized AgNPs at different concentrations (0.2-1.0mg/mL) was used for the analysis of DPPH scavenging activity of AgNPs and incorporated in the revised manuscript (L. 332-333).

Section 3.3.2:

Comment 18: L332-333: “The results of this study indicated that biosynthesized AgNPs exhibited maximum FRAP activity.” What do the authors mean here? The range is given as 22-47%, I would understand 100% as maximum activity, which is not reached. Please amend to clarify this.

Response: Section 3.3.2 Ferric reducing antioxidant power (FRAP) assay results has been rewritten in the revised manuscript (L. 344-348).

Section 4:

Comment 19: L408 & 414: The authors refer to “earlier reports”, please include citations.

Response: Thank you so much for this critical observation. We have now incorporated an appropriate below citation in the revised manuscript reference Raza et al. [41] (L.446).

Raza M.A.; Kanwal Z.; Rauf A.; Sabri A.N.; Riaz S.; Naseem S. Size- and shape-dependent antibacterial studies of silver nanoparticles synthesized by wet chemical routes. Nanomaterials. 2016, 74. doi: 10.3390/nano6040074.

Comment 20: L422-423: The order in which inhibition is reported is not correct, the most inhibited microbe is R. solanacearum with the least being B. subtilis. This should be rectified or reworded.

Response: We totally agree with your suggestion. Accordingly, the antibacterial activity of AgNPs rectified in the revised manuscript (L. 436-437).

Comment 21: L432 onwards: The authors at length discuss the previously reported properties of AgNPs. For the most part of this paragraph, it is unclear whether in those NP were green synthesised or not. This needs to be made very clear, as the relevance and comparison with the presented study and the discussion is critically based on this distinction. Response: TW appreciate the Reviewer for this comment and suggestion. To meet the Reviewer suggestion, the discussion part is now revised corrected and rectified in the revised manuscript.

Comment 22: L444: “captivates” should be changed to “captures”

Response: Thank you so much. As suggested, “captivates” has been replaced with “captures” in the revised manuscript (L 458).

Comment 23: L455-459: These sentences are incoherent. The manuscripts language quality deteriorates significantly towards the end, which should be addressed.

Response: The sentences have been re-written from the discussion section.

Comment 24: L493: The legend is incoherent.

Response: Sorry for this mistake. The legend is now revised for easy understanding (L512-513).

Section 5:

Comment 25: L496: “significant thrust” should be changed to “significantly growing”.

Response: Thank you so much for this suggestion. Accordingly, the word “significantly thrust” is replaced with “significantly growing” in the revised manuscript (L.515).

Comment 26: L504: What are the other biological activities the authors are referring to?

Response: We would like to express our special thanks to Reviewer for this comment. Biological activities refer toantifungal, antidiabetic, anti-inflammatory, cytotoxicity activities” (523-524).

Round 2

Reviewer 1 Report

Thank you to the Authors for clarifying questionable points. In my opinion, the amendments introduced are sufficient. I find that in this form, the article can be published.

Reviewer 2 Report

Thank you for the thorough correction of the manuscript and addressing and incorporating the suggested and requested changes.